# Prognostic Implication of Metastatic Lymph Node Ratio in Colorectal Cancers: Comparison Depending on Tumor Location

**DOI:** 10.3390/jcm8111812

**Published:** 2019-11-01

**Authors:** Jung-Soo Pyo, Young-Min Shin, Dong-Wook Kang

**Affiliations:** 1Department of Pathology, Eulji University Hospital, Eulji University School of Medicine, Daejeon 35233, Korea; 2Eulji University School of Medicine, Daejeon 34824, Korea

**Keywords:** metastatic lymph node ratio, colorectal cancer, survival, LN metastasis

## Abstract

Background: The proportion of the number of involved lymph nodes (LNs) to the number of examined LNs—defined as metastatic LN ratio (mLNR)—has been considered as a prognostic parameter. This study aims to elucidate the prognostic implication of the mLNR in colorectal cancer (CRC) according to the tumor location. Methods: We evaluated the correlation between prognoses and the involved and examined LNs as well as mLNR according to the tumor location in 266 surgically resected human CRCs. Besides, to evaluate the optimal cutoff for high and low mLNRs, we investigated the correlation between mLNR and survival according to the various cutoffs. Results: LN metastasis was found in 146 cases (54.9%), and colon and rectal cancers were found in 116 (79.5%) and 30 (20.5%) of the cases, respectively. The mean mLNRs were significantly higher in rectal cancer than in colon cancer (0.38 ± 0.28 vs. 0.21 ± 0.24, *P* = 0.003). Besides this, the number of involved LNs in rectal cancer was significantly high compared to colon cancer (11.83 ± 10.92 vs. 6.37 ± 7.78, *P* = 0.014). However, there was no significant difference in the examined LNs between the rectal and colon cancers (31.90 ± 12.28 vs. 36.60 ± 18.11, *P* = 0.181). In colon cancer, a high mLNR was significantly correlated with worse survival for all cutoffs (0.1, 0.2, 0.3, and 0.4). However, rectal cancer only showed a significant correlation between high mLNR and worse survival in the subgroup with a cutoff of 0.2. Conclusions: Our results showed that high mLNR was significantly correlated with worse survival. The number of involved LNs and mLNRs were significantly higher in rectal cancer than in colon cancer. The cutoff of 0.2 can be useful for the differentiation of prognostic groups, regardless of tumor location.

## 1. Introduction

Cancer staging is important for the stratification of a patient’s prognosis. Cancer staging, the primary tumor (pT), regional lymph node (pN), and distant metastasis (pM) were evaluated by the American Joint Committee on Cancer (AJCC) [1]. Among these parameters, the pN stage is only evaluated by the number of involved regional lymph nodes (LNs) in colorectal cancer (CRC) [1]. In CRC, a patient with regional LN metastasis is classified as stage III, in which adjuvant chemotherapy treatment is generally recommended [1,2,3]. Therefore, the detection of nodal status is essential for the decision of treatment modality and prediction of prognosis. The detection of LN involvement can be affected by various factors, including surgical and pathological status, as well as unexpected tumor conditions [4,5]. A previous study reported that only 37% of CRC patients performed adequate LN evaluation, and the number of examined LNs after surgical resection was significantly correlated with survival of the CRC patients [4]. The current guidelines recommended that 12 regional LNs be examined for proper evaluation of nodal disease [1,3,6]. However, the evaluation of examined LNs alone is not sufficient. Although the examined LNs have a prognostic role and could indirectly affect the involved LNs, the information is not conclusive. However, examined LNs are not considered in the current pN stage. An insufficient number of examined LNs may lead to a false-negative result for nodal disease or a lower N stage [7]. To compensate for these possibilities, many authors have introduced the metastatic lymph node ratio (mLNR) in examining various malignant tumors, such as gastric, pancreatic, breast, thyroid, cervix, and salivary gland cancers [8,9,10,11,12,13,14,15]. Berger et al. first reported the prognostic role of mLNR in CRC [16], though no detailed information regarding the criteria for high mLNR is currently available. Recently, the mLNR has been explored as a prognostic factor on survival outcomes and time to progression for patients with colon cancer [17,18,19,20,21,22,23,24,25,26]. 

This study aimed to elucidate the prognostic implications of mLNR in CRC. In addition, detailed information according to cutoff and tumor location was investigated using human CRC samples. The correlation between the involved and examined LNs was evaluated based on tumor location.

## 2. Materials and Methods

### 2.1. Patients and Evaluation of Pathological Features

A series of 266 patients (135 male and 131 female) who had undergone surgical resection of CRC at the Eulji University Medical Center between 1 January 2001 and 31 December 2010 were analyzed. The CRC specimens which received preoperative chemoradiotherapy were excluded from our study. Two independent authors reviewed medical charts, pathological reports, electronic operation records, and hematoxylin and eosin slides in order to assess clinicopathological characteristics such as age; sex; tumor size; tumor location; tumor differentiation; vascular, lymphatic, and perineural invasion; depth of tumor invasion; number of examined LNs; LN metastasis; distant metastasis; and pathologic tumor node metastatic (pTNM) stages. All cases were histologically confirmed as primary colorectal adenocarcinoma and evaluated according to the 8th edition of the AJCC Cancer Staging Manual [1]. The distant metastasis was indicated as the presence of cancer cells outside the area of surgical resection, such as lung, liver, pancreas, bone, and other organs. This protocol was reviewed and approved by the Institutional Review Board of Eulji University Hospital (Approval No. EMC 2018-11-005).

### 2.2. Definition and Evaluation of Metastatic Lymph Node Ratio

mLNR, which is defined as the ratio of the number of mLNs to the number of examined LNs, was calculated in CRCs with LN metastasis. To evaluate the optimal cutoff for high and low mLNR in the present study, the prognostic implication of various cutoffs, including 0.1 to 0.4, was compared. We categorized the values into high and low mLNRs according to the cutoffs and elucidated the prognostic implication of mLNRs.

### 2.3. Statistical Analysis

Statistical analyses were conducted using SPSS version 22.0 software (IBM Corp., Armonk, NY, USA). The correlation between LN metastasis and clinicopathological parameters was determined by either Pearson’s chi-square test or Fisher’s exact test (two-sided). Comparisons of the examined LN, involved LN, and mLNR between tumor locations were conducted using a two-tailed Student’s *t*-test. Linear regression analysis was used to investigate the correlations between the involved LN and examined LN. Recurrence-free survival (RFS) was defined as the duration from the operation date to the first date of recurrence or last follow-up date. In addition, overall survival (OS) was also indicated as the time from the date of surgery to the date of death or the last follow-up date, and the follow-up periods ranging from 0 to 60 months. Also, the prognostic implications of mLNR were evaluated by a Cox regression test. Results were considered statistically significant for *P* < 0.05.

## 3. Results

### 3.1. Clinicopathological Significance of Lymph Node Metastasis in Colorectal Cancers

Among 266 patients with CRC, LN metastasis was found in 146 patients (54.9%) with a mean age of 64.03. The correlations between LN metastasis and clinicopathological parameters in CRC patients are summarized in Table 1. 

The tumor size was significantly larger in patients with LN metastasis than in those without LN metastasis (5.69 ± 2.07 vs. 5.18 ± 2.05, *P* = 0.044). LN metastasis was frequently found in poorly differentiated cases in comparison to well or moderately differentiated cases. Also, there were significant correlations between LN metastasis and vascular, lymphatic, and perineural invasions. Patients with LN metastasis were significantly correlated with a higher pT stage, frequent distant metastasis, and a higher pTNM stage (*P* < 0.001, *P* = 0.001, and *P* < 0.001, respectively). No significant correlation was observed between the existence of LN metastasis and clinicopathological characteristics such as age, sex, location of the tumor, and the number of examined LNs.

### 3.2. Characteristics of Nodal Status in Colorectal Cancers

Of the 146 CRCs with LN metastasis, colon and rectal cancers made up 116 (79.5%) and 30 (20.5%) of the cases, respectively. The number of examined LNs showed no significant difference between colon and rectal cancers (36.60 ± 18.11 vs. 31.90 ± 12.28, *P* = 0.181). The mean number of involved LNs was significantly higher in rectal cancer than in colon cancer (11.83 ± 10.92 vs. 6.37 ± 7.78, *P* = 0.014). In addition, the mLNR was significantly higher in rectal cancer than in colon cancer (0.38 ± 0.28 vs. 0.21 ± 0.24, *P* = 0.003). The comparisons of the numbers of involved LN and mLNR based on tumor location and the correlation between mLN and tumor location in colorectal cancers are shown in Figure 1 and Table 2, respectively.

Next, the impact of examined LNs on involved LNs was evaluated in CRC. Summarized in Table 3 are the results of the correlation between the involved LNs and examined LNs in colorectal cancers by linear regression. Overall, the number of involved LNs increased with the increasing number of examined LNs (*P* = 0.037). Regarding tumor location, there was a positive correlation between involved LNs and examined LNs in the rectum, but not the colon (*P* = 0.023 vs. *P* = 0.068). 

### 3.3. Correlation between High Metastatic Lymph Node Ratio and Survival Rates in Colorectal Cancers

The optimal cutoffs for high and low mLNRs in CRC were obtained by predicting their roles in survival. In the present study, the evaluated cutoffs were 0.1, 0.2, 0.3, and 0.4. Overall, CRCs with a high mLNR were significantly correlated with worse OS and RFS for all cutoffs. The Kaplan-Meier survival analysis by the Cox regression test and the correlations between high mLNR and survival rates in colorectal cancers by Cox regression test are shown in Figure 2 and Table 4, respectively. In the subgroup analysis based on tumor location, there were significant correlations between high mLNR and worse survival in colon cancers. However, in rectal cancers, patients with high mLNR showed worse survival compared to those with low mLNR in cutoff 0.2, but not in other cutoffs. 

Besides, a high mLNR (≥0.2) was significantly correlated with tumor location, vascular and lymphatic invasions, perineural invasion, and the numbers of examined and involved LNs. The correlation between nodal status and survival rates in 146 CRCs with LN metastasis by Cox regression test and the correlation between mLNR and clinicopathological parameters in 146 CRCs with LN metastasis are summarized in Table 5 and Table 6, respectively.

## 4. Discussion

The most powerful prognostic factor for CRCs is an anatomical tumor extension, based on AJCC cancer staging [1]. This staging system is composed of evaluations for the extent of the primary tumor, the degree of spread to regional LN, and the presence of distant metastasis. In CRC, the pN stage is decided by the number of involved LNs. We focused on the mLNR for CRCs with regional LN metastasis. In addition, the present study evaluated the prognostic role of mLNRs according to the tumor location and the correlation between involved and examined LNs in the CRCs. The pN stage of CRC is divided into pN1 (one to three) or pN2 (four or more) by the number of involved LNs according to current AJCC cancer staging [1]. In CRC, the pN stage had prognostic implications; however, the pN stage is limited in terms of the delicate differentiation of patient prognoses. In our study, there was no significant difference in survival rates between patients at pN1b and pN2a stages (*P* = 0.970 and *P* = 0.483, respectively; data not shown). Besides, there was no significant difference in survival rates between patients with three and four involved LNs (*P* = 0.970 and *P* = 0.483, respectively; data not shown). Stratification of the same pN stage is needed due to the various prognoses. Therefore, the detailed evaluation of nodal status requires accurate prediction of the prognosis of CRC. The accuracy of the assessment must be ensured, as well as convenience. Although the present system has benefits due to the convenience of evaluation, additional parameters are needed for detailed stratification of a patient’s prognosis. They are necessary to assess the obvious usefulness of mLNRs, which have been studied. Given the convenience of the assessment tool, mLNRs, attained by simple calculation, can be useful in daily practice. In our results, there was a significant correlation between high mLNRs and worse survival in CRC. As the prognostic implications of mLNR are controversial between studies, detailed analyses are needed to examine mLNRs in CRC [17,18,19,20,21,22,23,27,28,29,30,31,32,33].

The present study investigated the involved and examined LNs to assess the impacts of mLNR in CRC. The number of examined LNs is important for the accuracy of the assessment using nodal status. In addition, the impact of the examined LN number on prognosis cannot be excluded in the evaluation of the prognostic role of mLNR. According to the guidelines, the guaranteed number is 12 LNs [1]. Some researchers have suggested that the minimum number of harvested LNs is 15 for evaluation using the ratio-based system [16]. In the current study, the mean number of examined LNs was 34.5 ± 18.2 in overall CRCs. Cases with an adequate number of LNs (≥12) totaled 94.7% (252 of 266 cases). However, in the previous study, the rate of CRC patients with adequate LN evaluation was only 37% [4]. The number of examined LNs may also be affected by the adequacy of the surgical resection and proper examining pathologists [1,34]. Indeed, the prognostic effect of the harvested LN number should be considered in the interpretation of the prognostic role of nodal disease. However, in AJCC cancer staging, pN is decided by only affected regional LNs in CRC. That is, a more comprehensive evaluation system for the nodal disease is needed. Previously, some researchers have reported on the prognostic role of examined LN [5,33]. Vather et al. reported the trend that increasing examined LNs were associated with increased mortality in stage III CRC [33]. However, they did not report statistical significance. In our results, the prognostic role of examined LNs was evaluated by dividing into two subgroups with high and low examined LNs (≥35 vs. <35). Although the subgroup with high examined LNs had prolonged survival rates, there was no significant difference in survival rates between the two subgroups. In stage III CRCs, the same results were obtained. Moreover, there was no significant difference between CRCs with and without LN metastasis in the examined LNs (35.64 ± 17.15 vs. 33.38 ± 19.38, *P* = 0.316). 

As described above, the number of involved LNs is only associated with the pN stage. The criterion—which is differentiated between pN1 and pN2—is four involved LNs. However, the proper number of examined LNs can be affected by the detected number of involved LNs [1]. In the present study, the impact of tumor location on the number of involved and examined LNs was investigated. Although the number of involved LNs increased according to the number of examined LNs (*P* = 0.037 in linear regression test, Table 3), this pattern was not consistent, depending on tumor location. In the subgroup analysis by tumor location, this finding was valid only in the rectum, but not in the colon (*P* = 0.023 and *P* = 0.068, respectively). Even though there was no significant difference in the number of examined LNs between the colon and rectal cancers, the number of involved LNs was significantly higher in rectal cancer. Therefore, the mLNR was significantly higher in rectal cancer than in colon cancer (*P* = 0.003). This finding could be caused by anatomical differences. These results suggest that different cutoffs are needed according to tumor location. 

The mLNR is defined as involved LNs divided by examined LNs. As described above, the mLNR can differ according to examined LNs. Thus, the mLNR can be useful for differentiating patients with the same number of involved LNs. For the application of mLNR in daily practice, evaluation of the cutoff value for high mLNRs is required. It is difficult to apply dichotomous data because it is not possible to confirm the significance of each mLNR value. However, dichotomous data may be more advantageous than continuous data for the application of mLNR. Previous studies used various criteria for high mLNR, from 0.125 to 0.3 [4,17,19,20,21,22,23,28,29,30,31,32,33,34]. In the present study, the prognostic impacts of high mLNR were investigated using various cutoffs (from 0.1 to 0.4). In addition, validation based on tumor location was conducted. For all cases, all cutoffs (0.1 to 0.4) allowed for the differentiation of the prognostic groups. However, results were discordant in the subgroup analysis based on tumor location. mLNR has been known as an independent prognostic factor in various malignant tumors, including the stomach, pancreas, breast, thyroid gland, cervix, and salivary glands [8,9,10,11,12,13,14,15]. However, in rectal cancers, some cutoffs had no prognostic role. The cutoff of 0.2 showed prognostic roles in both the OS and RFS of rectal cancers (*P* = 0.024 and *P* = 0.034, respectively, Table 4). In CRCs with LN metastasis, the mean numbers of involved and examined LNs were 7.5 ± 8.8 and 35.6 ± 17.1, respectively. The mean mLNR was 0.24 ± 0.26 in CRCs with LN metastasis. A high mLNR (cutoff ≥0.2) was not significantly correlated with pT or pM stage (*P* = 0.655 and *P* = 0.295, respectively); that is, in tumor progression, lymphatic invasion and LN metastasis can directly cause higher mLNR values. Interestingly, the mean mLNR was significantly higher in rectal cancers than in colon cancers (*P* = 0.003). However, there was no significant difference in the number of examined LNs between the colon and rectal cancers. From our results, it can be seen that mLNR may be more important in rectal cancer. In addition, in the evaluation of rectal cancers, more attention should be paid in cases with preoperative chemoradiotherapy, because the number of examined LNs can be significantly decreased after preoperative chemoradiotherapy [35,36]. Although the mLNR can be useful as a powerful predictor for prognosis of surgically resected CRCs, further detailed cumulative assessment is needed to confirm the optimal criteria of high mLNR.

## 5. Conclusions

Our results showed that high mLNR was significantly correlated with worse survival outcomes in CRC. Rectal cancers had a higher number of involved LNs, despite fewer examined LNs, compared to colon cancers. The cut-off of 0.2 can be useful for the differentiation of prognostic groups, regardless of tumor location in CRCs. For evaluation of optimal criteria of mLNR in CRCs, further comprehensive studies are required.

## Figures and Tables

**Figure 1 jcm-08-01812-f001:**
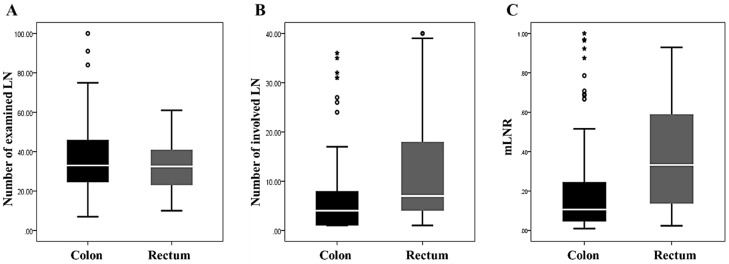
The comparisons of the numbers of examined and involved lymph nodes and metastatic lymph node ratio (mLNR) based on tumor location. (**A**) The number of examined lymph nodes (36.60 ± 18.11 vs. 31.90 ± 12.28, *P* = 0.181 in the colon and rectal cancers). (**B**) The number of involved lymph nodes (6.37 ± 7.78 vs. 11.83 ± 10.92, *P* = 0.014 in the colon and rectal cancers). (**C**) The mLNR (0.21 ± 0.24 vs. 0.38 ± 0.28, *P* = 0.003 in the colon and rectal cancers).

**Figure 2 jcm-08-01812-f002:**
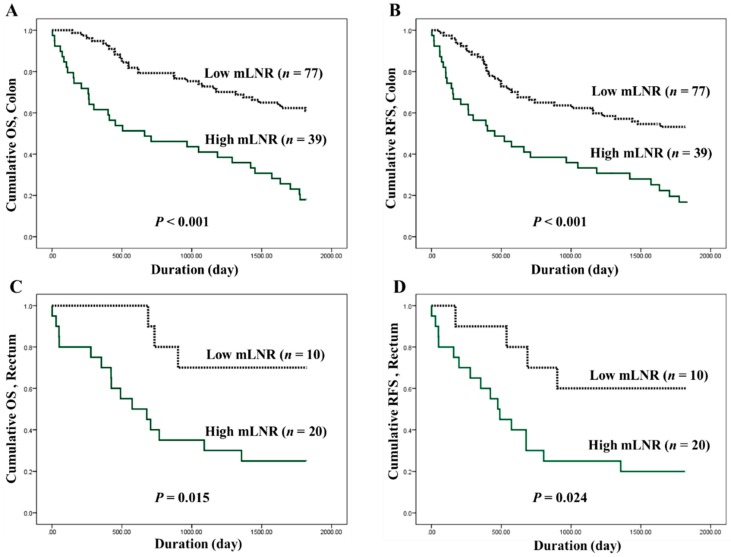
Kaplan-Meier survival curves between patients of low metastatic lymph node ratio (mLNR) (dotted line) and high mLNR (solid green line). (**A**) Cumulative overall survival (OS), and (**B**) recurrence-free survival (RFS) in 116 colon cancer patients. (**C**) Cumulative OS, and (**D**) RFS in 30 rectal cancer patients. The *P*-value was obtained using the log-rank test of the differences; *n* represents the patient’s number.

**Table 1 jcm-08-01812-t001:** The correlation between lymph node metastasis and clinicopathological parameters in colorectal cancers.

	Lymph Node Metastasis		*P*-Value
	Present	Absent	
Total (*n* = 266)	146 (54.9)	120 (45.1)	
Age (mean ± SD, years)	64.03 ± 13.42	63.04 ± 12.30	0.534
Sex			0.313
Male (*n* = 135)	70 (47.9)	65 (54.2)	
Female (*n* = 131)	76 (52.1)	55 (45.8)	
Tumor maximum diameter			0.192
≤5 cm	53 (36.3)	53 (44.2)	
>5 cm	93 (63.7)	67 (55.8)	
Tumor size (mean ± SD, cm)	5.69 ± 2.07	5.18 ± 2.05	***0.044***
Location of tumor			0.422
right colon	67 (45.9)	61 (50.8)	
left colon	79 (54.1)	59 (49.2)	
Tumor differentiation	108 (74.0)	103 (85.8)	***0.017***
Well or Moderate	38 (26.0)	17 (14.2)	
Poorly			
Vascular invasion			***<0.001*** ^+^
Present	22 (15.1)	2 (1.7)	
Absent	124 (84.9)	118 (98.3)	
Lymphatic invasion			***<0.001***
Present	57 (39.0)	13 (10.8)	
Absent	89 (61.0)	107 (89.2)	
Perineural invasion			***<0.001***
Present	35 (24.0)	9 (7.5)	
Absent	111 (76.0)	111 (92.5)	
Number of examined lymph node (mean ± SD)	35.64 ± 17.15	33.38 ± 19.38	0.316
pT stage			***<0.001***
pT1-2	9 (6.2)	32 (26.7)	
pT3-4	137 (93.8)	88 (73.3)	
Distant metastasis			***0.001***
Present	24 (16.4)	5 (4.2)	
Absent	122 (83.6)	115 (95.8)	
pTNM stage			***<0.001***
I–II	0 (0.0)	115 (95.8)	
III-IV	146 (100.0)	5 (4.2)	

*P*-values were estimated by Pearson’s chi-square test; ^+^*P*-values were estimated by Fisher’s exact test; numbers in parentheses represent percentage; SD, standard deviation; *P* < 0.05 are highlighted in bold italic.

**Table 2 jcm-08-01812-t002:** The correlation between metastatic lymph node and tumor location in colorectal cancers.

	Location of Tumor		*P*-Value
	Colon	Rectum	
Total (*n* = 146)	116 (79.5)	30 (20.5)	
Number of examined lymph nodes (mean ± SD)	36.60 ± 18.11	31.90 ± 12.28	0.181
Number of involved lymph nodes (mean ± SD)	6.37 ± 7.78	11.83 ± 10.92	***0.014***
Metastatic lymph node ratio (mean ± SD)	0.21 ± 0.24	0.38 ± 0.28	***0.003***

Numbers in parentheses represent percentage; SD, standard deviation, *P* < 0.05 are highlighted in bold italic.

**Table 3 jcm-08-01812-t003:** The correlation between involved lymph node and examined lymph node by linear regression in colorectal cancers.

Location of Tumor	R^2^	*P*-Value
Overall	0.030	***0.037***
Colon	0.029	0.068
Rectum	0.172	***0.023***

R^2^ represents the proportion of the involved lymph node for an examined lymph node in regression analysis; *P* < 0.05 are highlighted in bold italic.

**Table 4 jcm-08-01812-t004:** The correlation between metastatic lymph node ratio and survival rates in colorectal cancers by Cox regression test.

Metastatic Lymph Node Ratio	Overall Survival		Recurrence-Free Survival	
	Hazard Ratio	*P*-Value	Hazard Ratio	*P*-Value
**Overall**				
≥0.1 vs. <0.1	3.074 (1.864–5.069)	***<0.001***	2.559 (1.616–4.053)	***<0.001***
≥0.2 vs. <0.2	3.423 (2.186–5.361)	***<0.001***	2.751 (1.803–4.197)	***<0.001***
≥0.3 vs. <0.3	3.345 (2.136–5.237)	***<0.001***	2.822 (1.828–4.356)	***<0.001***
≥0.4 vs. <0.4	3.836 (2.375–6.195)	***<0.001***	3.233 (2.027–5.156)	***<0.001***
**Colon**				
≥0.1 vs. <0.1	2.972 (1.741–5.074)	***<0.001***	2.333 (1.423–3.826)	***0.001***
≥0.2 vs. <0.2	3.420 (2.071–5.649)	***<0.001***	2.701 (1.673–4.361)	***<0.001***
≥0.3 vs. <0.3	3.765 (2.221–6.382)	***<0.001***	3.395 (2.022–5.701)	***<0.001***
≥0.4 vs. <0.4	5.679 (3.131–10.302)	***<0.001***	5.141 (2.846–9.284)	***<0.001***
**Rectum**				
≥0.1 vs. <0.1	5.005 (0.664–37.706)	0.118	6.577 (0.876–49.385)	0.067
≥0.2 vs. <0.2	4.181 (1.203–14.533)	***0.024***	3.304 (1.095–9.968)	***0.034***
≥0.3 vs. <0.3	2.792 (1.040–7.497)	***0.042***	1.938 (0.789–4.764)	0.149
≥0.4 vs. <0.4	2.275 (0.890–5.814)	0.086	1.746 (0.723–4.219)	0.216

Numbers in parentheses represent 95% confidence intervals; *P* < 0.05 are highlighted in bold italic.

**Table 5 jcm-08-01812-t005:** The correlation between nodal status, lymphatic invasion, and survival rates in 146 colorectal cancers with lymph node metastasis by Cox regression test.

	Overall Survival		Recurrence-Free Survival	
	Hazard Ratio	*P*-Value	Hazard Ratio	*P*-Value
Lymph node metastasis	2.045 (1.398–2.990)	***<0.001***	2.193 (1.522–3.162)	***<0.001***
Lymphatic invasion	1.987 (1.367–2.889)	***<0.001***	1.775 (1.237–2.549)	***0.002***
pN stage (pN1/2 vs. pN0)	1.804 (1.457–2.233)	***<0.001***	1.849 (1.507–2.269)	***<0.001***

Numbers in parentheses represent 95% confidence intervals; *P* < 0.05 are highlighted in bold italic.

**Table 6 jcm-08-01812-t006:** The correlation between metastatic lymph node ratio and clinicopathological parameters in 146 colorectal cancers with lymph node metastasis.

	Metastatic Lymph Node Ratio		*P*-Value
	High	Low	
Total (*n* = 146)	60 (41.1)	86 (58.9)	
Age (mean ± SD, years)	64.32 ± 13.51	63.84 ± 13.43	0.832
Sex			0.923
Male (*n* = 70)	28 (47.5)	42 (48.3)	
Female (*n* = 76)	31 (52.5)	45 (51.7)	
Tumor maximum diameter			0.209
≤5 cm	25 (42.4)	28 (32.2)	
>5 cm	34 (57.6)	59 (67.8)	
Tumor size (mean ± SD, cm)	5.42 ± 2.01	5.88 ± 2.10	0.183
Location of tumor			***<0.001***
Right colon	16 (27.1)	51 (58.6)	
Left colon and rectum	43 (72.9)	36 (41.4)	
Tumor differentiation			0.074
Well or Moderate	39 (66.1)	69 (79.3)	
Poorly	20 (33.9)	18 (20.7)	
Vascular invasion			***<0.001***
Present	17 (28.8)	5 (5.7)	
Absent	42 (71.2)	82 (94.3)	
Lymphatic invasion			***<0.001***
Present	38 (64.4)	19 (21.8)	
Absent	21 (35.6)	68 (78.2)	
Perineural invasion			***0.007***
Present	21 (35.6)	14 (16.1)	
Absent	38 (64.4)	73 (83.9)	
Number of examined lymph node (mean ± SD)	30.60 ± 15.41	39.15 ± 17.50	***0.003***
Number of involved lymph node (mean ± SD)	14.30 ± 10.11	2.74 ± 2.12	***<0.001***
pT stage			0.655 ^+^
pT1-2	3 (5.1)	6 (6.9)	
pT3-4	56 (94.9)	81 (93.1)	
Distant metastasis			0.295
Present	12 (20.3)	12 (13.8)	
Absent	47 (79.7)	75 (86.2)	

*P*-values were estimated by Pearson’s chi-square test; ^+^*P*-values were estimated by Fisher’s exact test; Numbers in parentheses represent percentage; SD, standard deviation; *P* < 0.05 are highlighted in bold italic.

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
