# Peer review of "Prognostic Implication of Metastatic Lymph Node Ratio in Colorectal Cancers: Comparison Depending on Tumor Location"

_jcm, 2019, doi:10.3390/jcm8111812_

Round 1

Reviewer 1 Report

Congratulations on submiting an interesting article to the publication. The authors provided an interesting perspective on the issue of colorectal cancer survival. The data are well processed statistically, the conclusions are clear and useful. Maybe how long the data on patients have been collected. Over the past 10 years, ther have been many changes in colorectal cancer survival prediction, especially in genetics science. Neverthless, I think the article is suitable for publication.

Only one question : In table 5 are data that 1 patient is classifaied as stage I-II group with metastases in lymph nodes. Is it true?

Author Response

Reviewer 1.

Congratulations on submiting an interesting article to the publication. The authors provided an interesting perspective on the issue of colorectal cancer survival. The data are well processed statistically, the conclusions are clear and useful. Maybe how long the data on patients have been collected. Over the past 10 years, ther have been many changes in colorectal cancer survival prediction, especially in genetics science. Neverthless, I think the article is suitable for publication.

Only one question : In table 5 are data that 1 patient is classifaied as stage I-II group with metastases in lymph nodes. Is it true?

Response:

   As pointed by a reviewer, patients with stage I or II had no lymph node metastasis. We corrected the Table 1 and Table 6 (Table 5 before revise) and merged the duplicated data. We also fixed other unintended errors in our manuscript.

   Thank you for careful review.

Reviewer 2 Report

This is a scientific oncological/surgical report about the prognostic implication of metastatic lymph node ratio in colorectal cancers (Comparison depending on tumor Location).

Abstract: The suggested structure recommended by JCM for this section is not applied and the abstract immediately starts with the methods section. Statements in the abstract need to be clear and informative, for example, the following needs to be modified: “there was no significant difference in the examined LNs between the colon and rectal cancers” (total numbers and mean for colon and rectal cancers need to be clear).

There are typos, to be corrected.

1- Introduction: The background literature and study rationale are briefly mentioned and this section seems very short and needs to provide more and sufficient information about the LN involvement in colorectal cancer. Also, only 9 references are used in this section, with no reference from 2017, 2018 and 2019, the most recent one (Amin et al. 2016) is more than three years old.

More recent scientific papers should be used to improve this section. There are several typos Throughout the paper, in-text citations are not proper and need to be corrected.

2- Materials and methods:

Authors need to consider, discuss and justify the adequacy of the sample size for each group, in this study. Full and detailed medical, surgical, pathological and demographic data form the patients’ should be provided in full detail as appendices.

3- Results:

In Table 1, the caption needs to define the nature of numbers in the brackets. The caption for table 2 needs to be modified. Figures need to be revised and improved (both for unusual mistakes and the quality)

4- Discussion and Conclusion: the discussion and conclusions the authors draw are justified and connected with the broader argument made in the paper.

Author Response

Reviewer 2.

Abstract: The suggested structure recommended by JCM for this section is not applied and the abstract immediately starts with the methods section. Statements in the abstract need to be clear and informative, for example, the following needs to be modified: “there was no significant difference in the examined LNs between the colon and rectal cancers” (total numbers and mean for colon and rectal cancers need to be clear).

There are typos, to be corrected.

Response:

As suggested by a reviewer, we applied the style of abstract recommended by JCM. We added some details to make the statement clearly and informative in the manuscript. Changes or corrected sentences were shown in red color.

We underwent English editing by MDPI’s English editing service (ID: english-11365) for corrected use of grammar and standard medical terms. The certificate by MDPI’s English editing service is as bellow.

1- Introduction: The background literature and study rationale are briefly mentioned and this section seems very short and needs to provide more and sufficient information about the LN involvement in colorectal cancer. Also, only 9 references are used in this section, with no reference from 2017, 2018 and 2019, the most recent one (Amin et al. 2016) is more than three years old.

Response:

 As mentioned by a reviewer, we added sufficient information and background about the lymph node involvement in colorectal cancer and the other site's cancers, such as thyroid, cervix, and salivary gland cancers. We also added available references including the recent three years (total 36 references) and cited 26 references in the ‘Introduction’. Changes or added contents were shown in red color in the manuscript, and main contents are as follows:

“Therefore, the detection of nodal status is essential for the decision of treatment modality and prediction of prognosis. The detection of LN involvement can be affected by various factors, including surgical and pathological status, as well as unexpected tumor conditions [4,5]. A previous study reported that only 37% of CRC patients performed adequate LN evaluation, and the number of examed LN after surgical resection was correlated with survival of the CRC patients.[4]”

2- Materials and methods:

Authors need to consider, discuss and justify the adequacy of the sample size for each group, in this study. Full and detailed medical, surgical, pathological and demographic data form the patients’ should be provided in full detail as appendices.

Response:

We added sufficient information and sample size on CRC patients and the clinicopathological features. We tried to address the points raised by the reviewers as best as we can. Changes or added sentences were shown in red color in the manuscript and main contents are as follows:

“All cases were histologically confirmed as primary colorectal adenocarcinoma and evaluated according to the 8th edition of the AJCC Cancer Staging Manual.[1] The distant metastasis was indicated as the presence of cancer cells outside the area of surgical resection, such as lung, liver, pancreas, bone, and other organs.”

“Recurrence-free survival (RFS) was defined as the duration from the operation date to the first date of recurrence or last follow-up date. In addition, overall survival (OS) was also indicated as the time from the date of surgery to the date of death or the last follow-up date, resulting in a mean follow-up period was 65.0 months (ranging from 0 to 184.9 month)."

3- Results:

In Table 1, the caption needs to define the nature of numbers in the brackets. The caption for table 2 needs to be modified. Figures need to be revised and improved (both for unusual mistakes and the quality)

Response:

As mentioned by a reviewer, we correct the captions in Table 1, Table 2, and Table 6. We added a new Table 5 to provide more and sufficient information about nodal status, lymphatic invasion, and survival rates. We also revised Figure 1 and Figure 2, as well as improving the quality of the figures. Changes or added sentences were shown in red color in the manuscript, and revised figures and captions are as follows.

Figure 1. The comparisons of the numbers of examined and involved lymph nodes and metastatic lymph node ratio (mLNR) based on tumor location.

Figure 2. Kaplan-Meier survival curves between patients of low metastatic lymph node ratio (mLNR) (dotted line) and high mLNR (solid green line).

Table 5. The correlation between nodal status, lymphatic invasion, and survival rates in 146 colorectal cancers with lymph node metastasis by Cox-regression test

4- Discussion and Conclusion: the discussion and conclusions the authors draw are justified and connected with the broader argument made in the paper.

Response:

   As recommended by a reviewer, we added the content in the discussion and conclusions. We tried to address the points as best as we can. Changes were shown in red color in the manuscript.

Round 2

Reviewer 2 Report

Thank you for your consideration.